# Structure and mechanism of a tripartite ATP-independent periplasmic TRAP transporter

James S. Davies [1,2,13], Michael J. Currie [1,13], Rachel A. North [1,2,13] ✉,
Mariafrancesca Scalise [3], Joshua D. Wright[1], Jack M. Copping[4],
Daniela M. Remus[1], Ashutosh Gulati [2], Dustin R. Morado[5], Sam A. Jamieson [6],
Michael C. Newton-Vesty[1], Gayan S. Abeysekera[1], Subramanian Ramaswamy [7],
Rosmarie Friemann[8], Soichi Wakatsuki [9,10], Jane R. Allison [4],
Cesare Indiveri[3,11], David Drew[2], Peter D. Mace [6] & Renwick C. J. Dobson [1,12] ✉

In bacteria and archaea, tripartite ATP-independent periplasmic (TRAP) transporters uptake essential nutrients. TRAP transporters receive their substrates via a secreted soluble substrate-binding protein. How a sodium ion-driven secondary active transporter is strictly coupled to a substrate-binding protein is poorly understood. Here we report the cryo-EM structure of the sialic acid TRAP transporter SiaQM from *Photobacterium profundum* at 2.97 Å resolution. SiaM comprises a "transport" domain and a "scaffold" domain, with the transport domain consisting of helical hairpins as seen in the sodium ion-coupled elevator transporter VcINDY. The SiaQ protein forms intimate contacts with SiaM to extend the size of the scaffold domain, suggesting that TRAP transporters may operate as monomers, rather than the typically observed oligomers for elevator-type transporters. We identify the $Na^+$ and sialic acid binding sites in SiaM and demonstrate a strict dependence on the substrate-binding protein SiaP for uptake. We report the SiaP crystal structure that, together with docking studies, suggest the molecular basis for how sialic acid is delivered to the SiaQM transporter complex. We thus propose a model for substrate transport by TRAP proteins, which we describe herein as an 'elevator-with-an-operator' mechanism.

Transporter proteins play key roles in bacterial colonisation, pathogenesis and antimicrobial resistance[1–3]. Tripartite ATP-independent periplasmic (TRAP) transporters are a major class of secondary transporters found in bacteria and archaea—but, not in eukaryotes[4,5]. First reported over 25 years ago[6], they use energetically favourable cation gradients to drive the import of specific carboxylate- and sulfonate-containing nutrients against their concentration-gradient, including $C_4$-dicarboxylates, α-keto acids, aromatic substrates and amino acids[7].

A functional TRAP system is made up of a soluble substrate-binding 'P-subunit', and a membrane-bound complex comprising a small 'Q-subunit' and a large 'M-subunit'. For a small proportion of TRAP transporters, the Q- and M-subunits are fused into a single polypeptide[4,7–12]. TRAP transporters are different from almost all other secondary active transporters in that they can only accept substrates from the P-subunit[6,9,13–15]. Analogous to ABC importers, the P-subunit is secreted to capture host-derived substrates with high affinity and specificity. The substrate-loaded P-subunit subsequently delivers its cargo to the membrane transporter QM[7–10]. The best characterised TRAP transporters are those for sialic acid, otherwise known as the SiaPQM system[12–14]. Sialic acids are a family of nine-carbon sugars of which the most common is *N*-acetylneuraminate (Neu5Ac)[16]. Sialic acid TRAP transporters have a demonstrated role in bacterial virulence[14,17,18] and, as such, they represent an attractive class of proteins for the development of new antimicrobials against pathogenic bacteria that

use them during infection. Despite their potential as antimicrobial targets and classification as evolutionary-divergent sodium ion-driven transporters, the molecular basis of how TRAP transporters work is poorly understood, with low sequence identity to transporters with known structures.

In this work, we report the cryo-EM structure of the sialic acid TRAP transporter SiaQM from *Photobacterium profundum* at 2.97 Å resolution. Together with mutagenic and functional data, we identify the substrate and the sodium ion-binding sites. The structure suggests that TRAP transporters use an elevator-type mechanism and that they operate as monomers, rather than the typically observed oligomers for other elevator-type transporters. We report the SiaP crystal structure that, together with docking studies, suggest the molecular basis for how sialic acid is delivered to the SiaQM transporter complex. We propose a model for substrate transport by TRAP proteins, which we describe as an 'elevator-with-an-operator' mechanism.

## Results and discussion

### The cryo-EM structure of SiaQM

The SiaQM transporter complex from *P. profundum*[19] was selected for structural studies as it was found to be highly stable in detergent solution. To improve image alignment of the relatively small SiaQM complex for structural determination by cryo-EM[20], synthetic nanobodies against SiaQM were generated using a yeast surface-display platform[21]. Promising nanobodies were converted into larger megabodies, which ultimately led to the selection of the megabody $Mb_{Nb07}^{HopQ}$ (see 'Methods'). SiaQM is found to be a stable monomeric transporter complex (Supplementary Fig. 1a–c) and the size and shape is similar in either detergent (L-MNG) or amphipol (A8-35). The detergent purified SiaQM complex was exchanged into either amphipol or lipid nanodiscs and then combined with megabody to form the SiaQM-$Mb_{Nb07}^{HopQ}$ complex (Supplementary Fig. 1d). The sample preparation was next optimised for grid preparation, cryo-EM data acquisition and structural

determination (see 'Methods'). For the structure of SiaQM-$Mb_{Nb07}^{HopQ}$ in amphipol, final particles produced excellent 2D-classes with the best 3D reconstruction yielding an overall resolution of 2.97 Å (FSC = 0.143 criterion), extending to 2.2 Å in some regions (Supplementary Fig. 2, Supplementary Table 1). For the structure of SiaQM-$Mb_{Nb07}^{HopQ}$ in nanodiscs, an overall resolution of 3.03 Å was achieved (FSC = 0.143 criterion), also extending to 2.2 Å in the core of the protein (Supplementary Fig. 2, Supplementary Table 1). Overall, 580 out of 597 residues of the SiaQM complex could be built into both structures, as expected from the high-quality cryo-EM maps (Supplementary Fig. 3).

Both cryo-EM structures of the SiaQM complex reveal one copy of the Q-subunit and one copy of the M-subunit, and an overlay of these structures demonstrates they are essentially the same (r.m.s.d. of 0.5 Å). The megabody is bound on the extracellular side of SiaQM, making contacts with both the Q- and M-subunits (Fig. 1a, b). The M-subunit is made up of 12 TM segments with an $N_{Out}C_{Out}$ topology (Fig. 1c). The M-subunit shows the highest structural similarity to the sodium ion-coupled dicarboxylate transporter VcINDY (Dali server[22], Z-score = 25.7, sequence identity = 15%), which is a secondary active transporter operating by an elevator alternating-access mechanism[23]. Based on this structural comparison, we designate TMs 1-3 and TMs 7-9 as forming a "scaffold" domain and TMs 4-6 and TMs 10-12 as forming a "transport" domain. The transport domain consists of two structural-inverted repeats. The first repeat contains a helical hairpin TM4a and TM4b originating from the cytoplasmic side (HP$_{in}$), followed by the discontinuous helix TM5a-b and TM6. The second repeat has a helical hairpin from the periplasm side (HP$_{out}$), and is made up from TM10a and TM10b, TM11a-b and TM12. The two symmetry-related repeats superimpose with an r.m.s.d. of 2.2 Å over 77 Cα positions (using the amphipol solubilised structure). The scaffold domain is likewise made up from two structurally-inverted repeats, the first of which is formed by TM1, TM2 and TM3a-b, and the second by TM7, TM8 and TM9a (Fig. 1c). The scaffold structural-inverted repeats superimpose with an

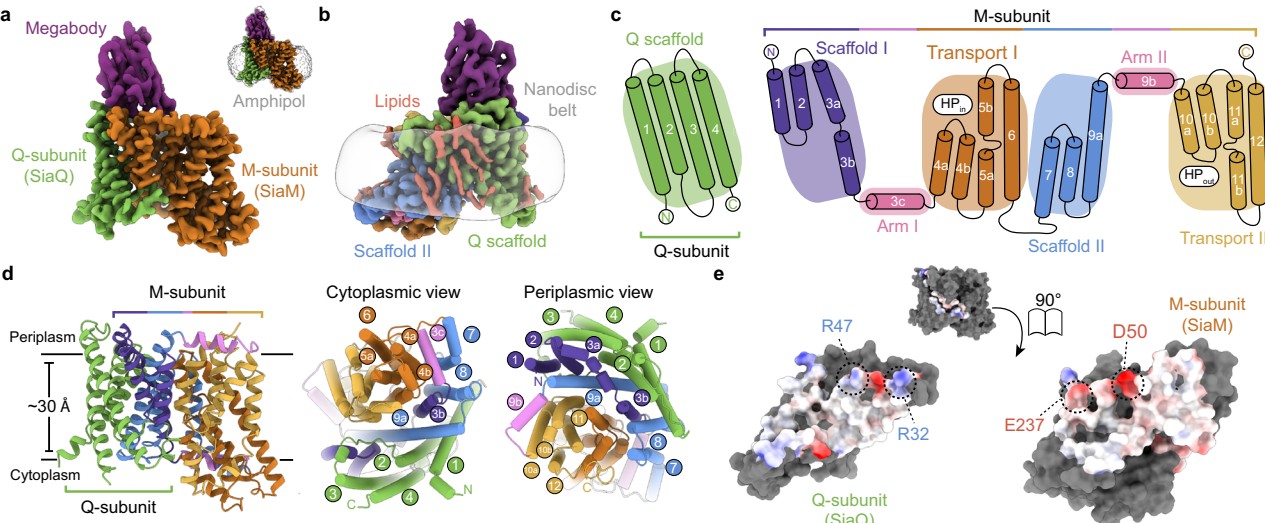

**Fig. 1 | Structure of *Photobacterium profundum* SiaQM. a** Cryo-EM density for the SiaQM-$Mb_{Nb07}^{HopQ}$ complex in amphipol. The megabody is bound at the periplasmic face of SiaQM and makes contacts with both the Q- and M-subunits. *Inset* shows density and position of the amphipol belt. Cryo-EM density displayed at 7.5σ, as calculated by *ChimeraX*[65]. **b** Cryo-EM density for the SiaQM-$Mb_{Nb07}^{HopQ}$ complex in MSP1D1-*E. coli* phospholipid nanodiscs (also displayed at 7.5σ). Density for lipids is shown in pink, while the contour of the disc is shown in grey. More detailed views are displayed in Supplementary Figs. 2 and 6. **c** Cartoon depicting the topology of SiaQM. The overall topology of SiaQM can be arranged into a transport domain and a rigid scaffold domain. The inverted topology of the M-subunit is seen in this cartoon, where Scaffold I, Arm I and Transport I form one repeat, with the remainder forming the second repeat. Hairpin helices HP$_{in}$ and HP$_{out}$ are indicated.

**d** Structural organisation of SiaQM. Left: SiaQM structure, showing the Q-subunit (green), and the M-subunit coloured by domain (scaffold domains: purple and blue; transport domains: orange and gold; arm helices: pink). Right: Views from the cytoplasm and periplasm, showing the scaffold domains (green, blue and purple) bracing the transport domains (orange and gold), which are cradled between the two arm helices (pink). **e** Surface representation of the interface between SiaQ and SiaM. The surfaces of the contact residues at the interface are coloured by electrostatic potential, as calculated in *ChimeraX*[65]. Highlighted are conserved residues that form salt bridges between the subunits, R32:E237 and R47:D50. *Consurf*[66] analyses show that R32 and E237 are fully conserved across the sequences sampled, highlighting the structural and functional significance of this pair.

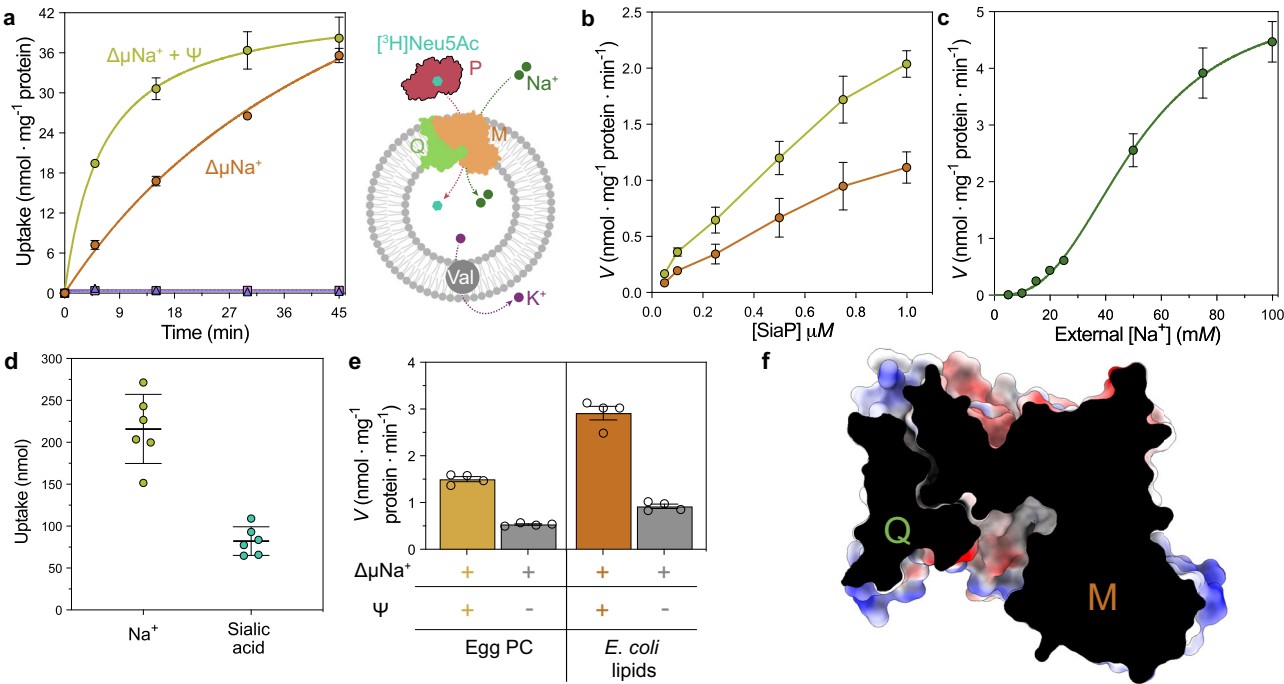

**Fig. 2 | Proteoliposome transport assays of SiaPQM. a** Transport by SiaPQM is dependent on an inwardly directed Na⁺ gradient and net transport is electrogenic as activity is enhanced when an inside negative membrane potential (ΔΨ, −117.1 mV) is imposed. Curves show external [³H]-Neu5Ac uptake into proteoliposomes reconstituted with SiaQM, in the presence of SiaP. Green circles, an inward sodium ion gradient (ΔμNa⁺) is present, with a membrane potential generated by valinomycin before measurement. Orange circle, ethanol was added instead of valinomycin as a control. Purple triangle, no ΔμNa⁺ was present, but a ΔΨ was imposed. Pink square, no ΔμNa⁺ was present and ethanol was added instead of valinomycin as a control. The plot presents the mean ± s.e.m. from five independent experiments (n = 5). **b** The dependence of Neu5Ac uptake into proteoliposomes based on SiaP. The conditions are the same as (**a**). SiaP is required for transport but the rate is not linear when the SiaP concentration is increased to 1.0 μM. All data are reported as means ± s.e.m. from four independent experiments (n = 4). **c** Dependence of Neu5Ac uptake into proteoliposomes based on external Na⁺ (green circle), with SiaP at 0.5 μM. Transport was measured in the presence of varying concentrations of external Na⁺-gluconate and fitted with the Hill equation, giving a Hill coefficient of 2.7 (95% CI = 1.9–4.0). The plot presents the mean ± s.e.m. from five independent experiments (n = 5). **d** Uptake of Na⁺, monitored by measuring the fluorescence emission of Sodium Green™, versus the uptake of [³H]-Neu5Ac into proteoliposomes. The mean Na⁺ uptake was 216 ± 41 nmol, while the mean sialic acid uptake was 82 ± 17 nmol (two-sided T-test P = <0.0001, determined using GraphPad Prism), giving a ratio of 2.6, consistent with the Hill coefficient in (**c**). The plot presents the mean ± s.d. from six independent experiments (n = 6) in each. **e** SiaQM transport activity is sensitive to the lipid environment. Transport was measured using proteoliposomes reconstituted with phosphatidylcholine from egg-yolk or *E. coli* total lipid extract. As a ΔΨ control, ethanol was used instead of valinomycin (orange bars). The bar presents the mean ± s.e.m. from four independent experiments (n = 4) in each. **f** Surface cutaway of SiaQM. The structure is in an inward-facing conformation, which is the substrate release state, with the presence of a large solvent-accessible cavity on the cytoplasmic face of the complex. Source data are provided as a Source data file.

r.m.s.d. of 3.8 Å over 80 Cα positions and are connected to their neighbouring transport domains by lateral 'arm helices' that cradle the transport domain (Fig. 1c). The arm I helix appears to be locked in place by a cation-π interaction formed between the conserved R75 in TM3c and Y254 in TM8 of the scaffold I domain (Supplementary Figs. 4 and 5). The arm II helix 9b also appears to be stabilised by a cation-π interaction to TM9a in the scaffold II domain (Supplementary Fig. 5). At the opposite end of each helix, where the arms join the transport domain, the loop regions are longer and more flexible, supporting that the transport domain can move independently of the scaffold domain.

The Q-subunit has an NInCIn topology as previously proposed[24,25]. It comprises three long helices (TM1, TM3 and TM4) and one shorter helix (TM2) with an extended cytoplasmic linker to TM3, and is positioned at an oblique angle (~40°) relative to the membrane normal (Fig. 1d). The Q-subunit extensively interacts with the scaffold domain of the M-subunit, burying a total surface area of ~2400 Å². SiaQ and SiaM interactions are dominated by van der Waals contacts, in addition to highly-conserved residues forming two salt bridges (Fig. 1e). The Q-subunit is enriched with tryptophan residues at the phospholipid-water interface (Supplementary Fig. 4), located near the termini of membrane spanning helices TM1, TM2 and TM4. This enrichment is found in other TRAP transporter sequences and in the scaffold domains of DASS family transporters (Supplementary Table 2). In an elevator mechanism, the substrate is only translocated by the transport domain that moves against the scaffold domain, which is fixed due to oligomerisation[26]. Given the extensive interaction of the Q-subunit with only the scaffold domain of the M-subunit and the known role of tryptophan residues for anchoring helices in membranes[27], it seems likely the role of the Q-subunit is for extending the "scaffold" domain so that SiaQM can function as a monomer. Indeed, in the SiaQM structure solved in nanodiscs, lipid-like densities are visible perpendicular to the membrane plane, with two distinctive densities adhering to TMs of the Q-subunit (Fig. 1b, pink) and adjacent to a number of tryptophan residues (Supplementary Fig. 6).

## SiaQM is electrogenic and SiaP dependent

To confirm the transport properties SiaPQM from *P. profundum* are equivalent to SiaPQM from *Haemophilus influenzae*[15], purified SiaQM was reconstituted into liposomes for transport assays. Uptake of ³[H]-Neu5Ac into SiaQM containing liposomes is strictly dependent on both the presence of an inwardly directed Na⁺ gradient, and the presence of the soluble substrate-binding protein SiaP (Fig. 2a, b). Net transport by SiaQM is electrogenic, as activity is enhanced when an inside negative membrane potential (ΔΨ) is imposed (Fig. 2a). Since Neu5Ac has a

single negative charge at neutral pH, electrogenic transport means at least two Na$^+$ ions are transported for every sialic acid molecule imported. Varying the external Na$^+$ at a fixed substrate concentration gives rise to a Hill coefficient of 2.7, implying two or more Na$^+$ ions are co-transported during each transport cycle (Fig. 2c). To corroborate this finding, the concentration of both Na$^+$ ions and co-transported sialic acid was directly measured using a fluorometric assay (using Sodium Green™ dye[28]) and the transport assay (using radiolabelled [$^3$H]-sialic acid). From the comparison (Fig. 2d), a ratio of 2.6 Na$^+$ ions per sialic acid was derived, which correlates well with the co-transport of two or more Na$^+$ ions.

In several elevator transporters, lipids have been shown to inter-calate between the scaffold and transport domains, where specific lipid interactions may help to grease structural transitions[26,29,30]. We note that SiaQM transport activity is increased two-fold when pro-teoliposomes were prepared using *Escherichia coli* phospholipid extract containing 70% of the non-bilayer forming lipid phosphatidy-lethanolamine, as compared with proteoliposomes prepared using phosphatidylcholine (Fig. 2e). This increase could be influenced by an altered reconstitution efficiency, but is consistent with our observa-tions of distinctive lipid-like densities found at the scaffold, and also at the interface between the rigid arm I helix and the transport domain (Fig. 1b, pink density).

## Substrate and sodium ion sites in SiaQM

The SiaQM structure is in an inward-facing conformation with a large solvent-accessible cavity of 793 Å$^3$ facing towards the cytoplasm (Fig. 2f). We identified two sodium ion sites in the SiaQM structure (Supplementary Fig. 7a). The first sodium ion site, Na1, is located between the loop of HP$_{in}$ and the unwound region of TM5. The cryo-EM map shows density that we attribute to a Na$^+$ ion coordinated in a trigonal bipyramidal pattern by the five backbone carbonyls of S103, S106, G145, V148 and P150. We observe a second sodium ion site, Na2, located between the loop of HP$_{out}$ and the loop between TM11a and TM11b. This second sodium ion site, Na2, is not as well-defined, but we identify that the backbone carbonyls of residues G325, G366, T369 and M372, and the sidechain hydroxyl of T369 are positioned for Na$^+$ coordination. At each of the defined sodium ion sites is a highly-conserved twin proline motif located at the peptide break of TM5a-b and TM11a-b, which was hypothesised to be required for sodium ion site formation[24]. Our structure confirms these prolines are juxtaposed at both Na$^+$ ion-binding sites and match those in the VcINDY structure, where the Na$^+$ ions are positioned either side of the substrate[23,31] (Sup-plementary Fig. 7b). Indeed, a surface topography analysis of the transport domain using CASTp[32] identified a pocket with a solvent accessible volume of 79 Å$^3$, which forms part of the large 793 Å$^3$ vesti-bule (Supplementary Fig. 7a), which is where we suggest that sialic acid binds. As also seen in VcINDY, the coordination of Na$^+$ ions are strengthened by the helix dipole moments of Hp$_{in}$ and TM5a for Na1, as well as Hp$_{out}$ and TM11a for Na2. We confirm the functional importance of these two sodium ion sites using the liposome assay, where the mutants S106-S108A (Na1) and T369A (Na2) show an almost com-plete knockdown of transport activity (~3% of wild type) (Fig. 3b). Unlike other unrelated sodium ion-coupled sialic acid transporters that adopt the LeuT-fold[33,34], the substrate binding-site lacks positively-charged residues for coordinating the negatively-charged sugar. Rather, the electrostatic surface potential shows a slight positive charge on one side of the cavity (above HP$_{in}$), with a stronger negative charge from E329 opposite (below HP$_{out}$) (Supplementary Fig. 7a), which is the only titratable sidechain in the binding cavity. Across all sialic acid TRAPs surveyed, this position is either a glutamic or aspartic acid (Supple-mentary Fig. 4). Mutation at this site (Fig. 3b) led to abrogation of transport activity, suggesting this residue is involved in substrate coordination. This fits with the binding mode seen in the crystal structure of SiaP (Supplementary Fig. 8a–c), where conserved acidic

residues are within hydrogen-bonding distances with hydroxyls on the glycerol tail of sialic acid. Despite sample preparation in the presence of 10 mM Neu5Ac for cryo-EM data collection, there is no detectable density for the substrate. Based on our structure, it seems likely a conformational change in SiaM is required to accept the substrate, which is also consistent with the requirement of SiaP for transport.

## Coupling of SiaP to SiaQM

We determined the crystal structure of SiaP bound to Neu5Ac in a closed state at 1.04 Å resolution, as a crystal structure of substrate-loaded SiaP enables us to model the SiaPQM complex (Supplementary Fig. 8, Supplementary Table 3). The algorithms *RaptorX*[35], *Gremlin*[36] and *AlphaFold*[37,38] were found to predict similar contacts, where the P-subunit interacts with both the Q- and M-subunits (Fig. 3a). The periplasmic surface of SiaQM shows a bowl-like shape with a large area where SiaP docks. The bowl is lopsided, with the lip at the side of the Q-subunit higher than that of the M-subunit. Predicted contacts involve mainly surface residues of the scaffold domain and can be used to orient the P-subunit with respect to the Q- and M-subunits (Fig. 3a).

In the SiaP structure, Neu5Ac is situated in a deep cleft bound by multiple residues, including R145, which is highly conserved in TRAP transporter substrate-binding proteins[7,8]. We confirmed that SiaP binds sialic acid with nanomolar affinity and that the crystal structure is representative of the protein structure in solution (Supplementary Fig. 7a). The docked structure shows two calliper-like helices of SiaP, α2 at the N-terminal lobe and α5 at the C-terminal lobe of SiaP, interact with the scaffold portion of the M-subunit (Supplementary Fig. 7a). Specifically, TM3a aligns well with α2 of the P-subunit, as does the loop between TM7 and TM8 with α5 of the P-subunit (Fig. 3a). Mutation of predicted interacting residues at these sites validate this tripartite model—R49A (SiaP, α2) and D50A (SiaM, TM3a) as well as the con-served E170A (SiaP, α5) result in a significant reduction in transport activity (Fig. 3b) (temperature stability, sialic acid binding and purity of all SiaP and SiaQM mutants are reported in Supplementary Fig. 11). The two periplasmic loops of the Q-subunit are also predicted to interact with the P-subunit, where there are high levels of sequence conserva-tion and surface charge complementarity at the interacting surfaces (Fig. 3c). Contacts from the transport domains are localised to the short helix TM5b in the Transport I domain and the loop connecting TM5b to TM6, which has predicted contacts with SiaP at a short 3$_{10}$ helix (η5) spanning residues F195 to E197.

Next, we used the docked tripartite model to predict how Neu5Ac is delivered to SiaQM for transport. We observe regions of surface and charge complementarity, sequence conservation and a high co-evolution signal, consistent with the docking and allosteric opening of SiaP with SiaQM. The predicted binding mode is similar to the mode first predicted by Ovchinnikov et al.[36], as well as that suggested for the *Hi*SiaPQM system. In our model, we define hotspots involving potential salt-bridges are formed between the scaffold domain and the N- and C-terminal lobes of SiaP, likely contributing to both recognition and allosteric modulation of the P-subunit (Fig. 3d). Mutations at these hotspots result in clear reductions in transport activity. At one hotspot, where R292 (SiaM) and D59 (SiaP) form a putative salt bridge in the model, R292A (with wildtype SiaP) gives an ~87% reduction in activity and D59A (with wildtype SiaQM) gives a ~77% reduction (Fig. 3b, d). We note that R292 is involved in the cation-π interaction that appears to rigidify one end of arm II (Supplementary Fig. 5). This arm helix links the scaffold to the transport domain and is likely important for the recognition and docking of SiaP and potentially for conformational transitions of SiaM. This finding is supported by recent work by Peter et al., who show that mutation of the equivalent residues in *Hi*SiaPQM (R484 and S60) affects growth in an *E. coli* complementation experiment[39].

We further identify two key hotspots from the model that are significant in that they present connections from the interacting

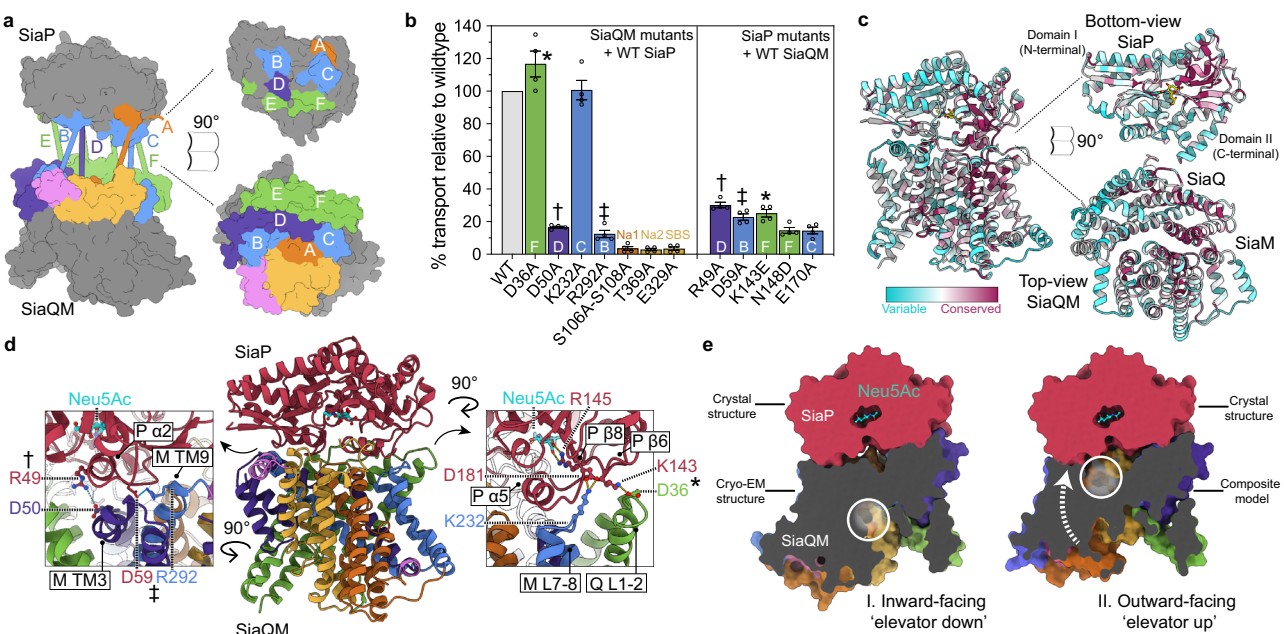

**Fig. 3 | Predicted full TRAP complex. a** Interacting regions of SiaQM and SiaP, as determined using the algorithms *RaptorX*[35], *Gremlin*[36] (Supplementary Tables 4 and 5) and *AlphaFold*[37]. Surface representation of the SiaQM and SiaP structures with patches coloured and lettered on each surface to indicate the binding mode. The proposed binding surface on SiaQM largely involves the surface of the scaffold (green, purple and blue). TM3a aligns well with α2 of the P-subunit (B blue and D purple). The loop between TM7 and TM8 aligns well with α5 of the P-subunit (C blue). The two periplasmic loops of the Q-subunit are also predicted to interact with the P-subunit (E and F, green). The orange and pink patches represent the Transport I and Arm II components as coloured in Fig. 1b. **b** Mutagenesis of surface residues on both SiaP and SiaQM that are predicted to be important for function, as well as residues in the SiaM substrate and Na⁺ binding sites. The data are normalised to the wildtype SiaPQM transport rate (grey). Mutants are coloured according to the regions shown in Fig. 1a. Data are reported as means ± s.e.m. (error bars) from four independent experiments. Source data are provided as a Source data file. Symbols mark residues pairs depicted in (**d**). **c** Residue conservation mapped onto the SiaP and SiaQM structures. Cartoon representation of SiaP and SiaQM, coloured according to *Consurf* score[66]. **d** Complex of the SiaQM cryo-EM structure and SiaP crystal structure based on the binding mode predicted by *AlphaFold*[38,67] (the final model can be found in Supplementary Data 1). An interaction hotspot (inset, left and right) shows a number of titratable residues at the interface. **e** Modelling of the TRAP complex in the inward-facing (left) and outward-facing (right) conformations (cutaway, viewed from the scaffold domain, looking towards the substrate-binding site, white circle). Left, the experimental structures determined here are aligned based on *AlphaFold* predicted complexes. Right, an outward-facing model was generated by homology modelling the SiaM transport domain using LaINDY (PDB ID: 6wu4) as the template (the final model can be found in Supplementary Data 2). This model was then aligned to the scaffold domain of model I. Together, these models demonstrate how an elevator motion of the transporter could expose the substrate binding site of SiaM to SiaP.

surfaces to the substrate-binding site of SiaP. One hotspot involves the invariant D181 of SiaP immediately prior to β8 and K232 on the loop between TM7 and TM8 (L7-8) of the M-subunit (Fig. 3b, d, right). Adjacent to this interaction, the surface-accessible, conserved K143 on the β6 strand of SiaP interacts with D36 on the loop between TM1 and TM2 of the Q-subunit (Fig. 3b, d, right). This strand in SiaP extends from the protein surface to the Neu5Ac binding site, with K143 located two residues upstream of the conserved R145 which directly coordinates Neu5Ac. Thus, SiaP binding to SiaQM may pull on this strand to release the substrate. Analysis of these hotspots using mutagenesis showed that K143E caused a significant reduction in transport activity (but not the ability of SiaP to bind sialic acid, Supplementary Fig. 11a), though the putative interactors D36A (SiaQ) and K232A (SiaM) did not reduce transport compared to the wildtype. Interestingly, mutation of the highly conserved D181 (SiaP) was not tolerated and the protein could not be purified (Supplementary Table 6). These data together suggest that this hotspot is important, yet the interactions are certainly more complex than the model predicts.

Residues R49 and N148 (SiaP) form a latch between the lobes of SiaP at the surface of the substrate binding cleft, enclosing a number of ordered solvent molecules that extend toward the substrate. Residues of TM3a (SiaM) have high co-evolution signal with α2 of SiaP (Supplementary Table 5) and form another interacting hotspot. Here, D50 (SiaM, TM3a) is well-poised to interact with the aforementioned latch. It has already been established that changes at the interacting surface of SiaP disrupts the ordered water molecules in the binding cleft and

alters the affinity of SiaP for sialic acid[40]. R49A (latch-disrupting), N148D (latch-strengthening) and D50A (potential interactor from SiaM) mutants all show a reduction in transport activity (Fig. 3b), demonstrating the significance of this area for SiaQM function and highlighting a mechanism by which the nanomolar affinity of SiaP for sialic acid may be modulated by surface interactions with SiaQM. We note that in the cryo-EM structures, D50 is involved in an interaction between SiaQ and SiaM (Fig. 1), but the D50A mutant did not disrupt assembly of SiaQM and the complex was stable throughout purification. Analysis of this region by Peter et al. only showed a minor growth defect with N150D in an in vivo assay, perhaps suggesting this mutation is not strong enough to fully lock the SiaP in a closed conformation[39]. Finally, at this central interaction point, mutation of the conserved E170A (SiaP, α5) results in greatly reduced activity (Fig. 3b). Here, potential interactors on SiaM are less clear, although there is a patch of positive charged residues on the corresponding surface of TM3a-TM3b adjacent to D50. This result is supported by a charge swap mutation of the equivalent residue in *Hi*SiaP (E172) to an arginine, which gives a significant growth defect[39]. Together, the tripartite model and functional data suggest that multiple sites play a role in the recognition and allosteric interactions necessary for transport.

## An elevator-with-an-operator mechanism

The M-subunit shares a conserved fold with divalent anion sodium symporter (DASS) family members VcINDY, LaINDY and NaCT, as well as the bacterial AbgT-type transporter[41] (Supplementary Fig. 9). These

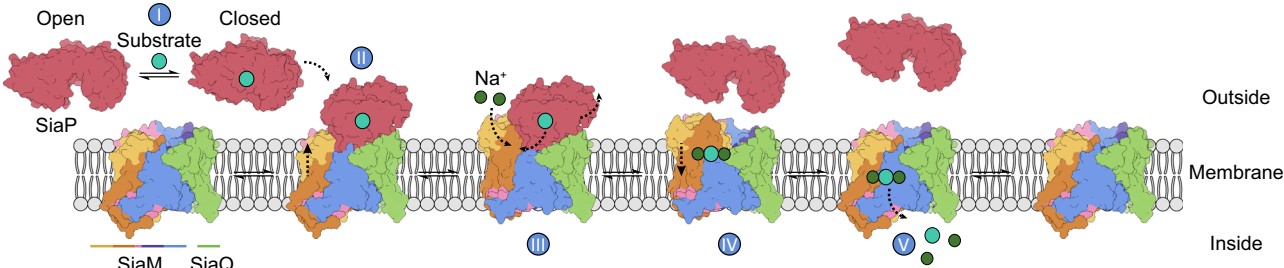

**Fig. 4 | The TRAP 'elevator-with-an-operator' mechanism.** (I) The P-subunit (maroon) binds the substrate (cyan) with high affinity and undergoes a conformational change from the open to closed state. (II) The closed P-subunit then docks to the QM subunits (orange, purple and green). (III) We propose that docking induces a conformational change in the transporter to a state where Na⁺ ions (green) and the substrate can bind with greater affinity. This change is coupled to the allosteric modulation of the P-subunit to the open conformation, releasing the substrate to the transporter. (IV) The open state P-subunit, which presumably has lower affinity for the transporter, diffuses away, allowing the transporter to move to an inward-facing state (V), with the substrate and coupling ions then released into the cytoplasm. We note it is possible the conformational change induced in the transporter (II) may be either a local gating rearrangement, or a global elevator-type motion. Regardless, we suggest that the P-subunit is the 'operator' of the elevator, as transport without the P-subunit is negligible, as seen in Fig. 2b.

transporters undergo elevator-structural transitions, which involve a vertical translation and a rigid-body rotation of the transport domain against the scaffold[42]. Molecular dynamics simulations of LaINDY show that elevator-like motions induce the bending of flexible loops connected to the arm helices and the hairpin helices in the transport domains[42]. Due to a shared fold and the rigidity of the arms in our SiaQM structure, we suggest TRAP transporters undergo similar conformational changes. We have modelled the outward-facing 'elevator up' state of the TRAP using LaINDY (currently the only DASS-type transporter solved in an outward-facing state) as a homology model for the transport domain (Fig. 3e, model on right) and compared this to the 'elevator down' state (see Supplementary Fig. 12 for further detail on the proposed motion of the transport domain). This model is consistent with the TRAP being an elevator-type transporter and shows movement of the SiaM binding site towards the surface of the SiaP binding cleft when overlaid with the original tripartite model. The elevator movement of the transport domain does not produce significant clashes with SiaP in the tripartite model and is further supported by analysis of coevolved residues—the elevator motion brings areas with high co-evolution signal together, that were previously not in contact in the original 'elevator down' tripartite model, such as L5b-6 of SiaM and η5 of SiaP (Supplementary Table 5). Similar to the DASS members, there are very few interactions between the transport and scaffold domains, further supporting our proposal that the M-subunit moves as a rigid body independently against the scaffold domain. Here, it seems that the extra scaffolding provided by the Q-subunit enables the transporter to uniquely function as a monomeric elevator. It is possible that the larger scaffold may have arisen to accommodate the interaction with SiaP, which could have become hindered upon an oligomeric assembly of SiaM. This motion and mechanism are consistent with that proposed by Peter et al.[39], who similarly used a model of VcINDY in the outward state. Here the authors also present a 4.7 Å cryo-EM structure of the fused HiSiaQM system. Our structures confirm the reported fold (see Supplementary Fig. 12 for an overlay), and provide further detail at the residue level, allowing us to identify ion and lipid binding sites and resolve the loop regions.

Although the transporter component of TRAP transporters shares no sequence or structural similarities with ABC transporters, the mechanism of substrate translocation shares elements with Type I ABC importers[43]. Essentially, the P-subunit is analogous to the substrate-binding protein of ABC importers, which bind substrates with very high affinity and specificity. The substrate-binding protein then passes the substrate to the transporter, which otherwise has no, or poor, affinity for the substrate on its own. The consistent observation that substrate translocation requires the P-subunit is strong evidence that the substrate-binding protein modulates the SiaQM structure.

Critically, the predicted binding interaction mode between the P-subunit and SiaQM is over the scaffold domain. Almost certainly, the binding of SiaP triggers a conformational change in SiaQM that will enable the transport domain to accept the substrate, although it is unclear whether this is a local gating rearrangement and/or a global conformational switch from an inward- to an outward-facing conformation. Subsequently, substrate-free SiaP releases its interaction with SiaQM, which enables SiaP to re-capture another substrate molecule. We propose that the TRAP transport cycle can be described as an 'elevator-with-an-operator' mechanism, as modelled in Fig. 4, although the order of events (e.g., when Na⁺ ions bind) is yet to be elucidated. We speculate that Na⁺ binding occurs prior to substrate coordination, helping to correctly organise the substrate binding site, as recently proposed for VcINDY[44]. Coupling between sodium ion and substrate binding seems likely given the proximity of Na1 and Na2 to the substrate binding site in the structure. Our proposed mechanism is supported by the tripartite model, modelling of the outward-facing conformation, as well as mutagenesis at the interacting surfaces. The elevator-with-an-operator mechanism makes the reverse pathway (export) unfavourable—consistent with a previous observation that unphysiologically high concentrations of unliganded SiaP are required for the transporter to run in reverse[15]. This mechanism is therefore in line with the function of TRAP transporters, that is, to capture scarce nutrients with high affinity. In conclusion, our work highlights how the transport cycle of a small molecule transporter can be controlled by the donation of the substrate from a secreted protein, greatly expanding our general view of secondary active transport.

## Methods

### Protein expression and purification

The sequence encoding the genetically non-fused *Photobacterium profundum* SiaQM (Supplementary Table 7) was cloned into pBAD-HisA (GeneArt) using XhoI and EcoRI restriction sites so that a hex-ahistidine affinity tag was fused at its 5′ end. The sequence retained the native intergenic region (5′-GGATTTTTC-3′) between the Q- and the M-subunits. This plasmid was transformed into *Escherichia coli* TOP10 cells (Invitrogen). Cells were grown at 37 °C to log phase at an $OD_{600}$ of 1.7–1.9 in Terrific-Broth (TB). Recombinant protein expression was induced with 0.2% arabinose for 3 h. Cells were collected and homogenised in phosphate-buffered saline (PBS) pH 7.4, 0.5 mg mL⁻¹ lysozyme, 0.1 mM PMSF and lysed by ultrasonication at 70% amplitude in 0.5 s on, 0.5 s off cycles for 30 min, using a Hielscher UP200S Ultrasonic Processor. Cell debris was removed by two centrifugation steps at $16,000 \times g$ and 4 °C for 30 min. Membranes were harvested by ultracentrifugation at $240,000 \times g$ in a 50.2 Ti rotor (Beckman Coulter) at 4 °C for 2 h, and solubilised in PBS pH 7.4, 6% glycerol, 5 mM DTT,

0.1 mM PMSF and 2% w/v lauryl maltose neopentyl glycol (L-MNG, Anatrace) at 4 °C for 2 h. Insoluble material was removed by ultra-centrifugation at 160,000 × g in a 50.2 Ti rotor (Beckman Coulter) at 4 °C for 1 h. SiaQM was purified by Ni$^{2+}$-NTA affinity using a 5 mL HisTrap HP column equilibrated in 50 mM Tris-HCl pH 8.0, 150 mM NaCl and 0.002% (w/v) L-MNG at 4 °C. Solubilised material was loaded onto the column and washed with 20 column volumes (CVs) of equi-libration buffer, and bound protein was eluted in 10 CVs of equilibration buffer supplemented with 500 mM imidazole. Eluted protein was concentrated for size-exclusion chromatography (SEC) using 100 kDa molecular weight cut-off (MWCO) spin concentrators (Pall). SEC was carried out using a HiLoad 16/60 Superdex 200 size-exclusion column (Cytiva) in buffer comprised of 50 mM Tris-HCl pH 8.0, 150 mM NaCl and 0.002% w/v L-MNG at 4 °C. SEC fractions containing SiaQM were pooled and concentrated. Protein was either used in subsequent experiments, or flash-frozen in liquid nitrogen and stored at −80 °C for future use.

The sequence encoding SiaP without the native signal peptide (residues 1–22, from SignalP[45]) (Supplementary Table 7) was cloned into pET30ΔSE and transformed into *E. coli* BL21(DE3). Cells were grown at 37 °C to log phase at an OD$_{600}$ of 0.6–0.8 in Luria-Broth (LB) or an OD$_{600}$ of 0.3 in M9 minimal medium for isothermal titration calorimetry (ITC) experiments. Recombinant protein expression was induced with IPTG (1 mM) for 16 h at 26 °C. Cells were collected and lysed by ultrasonication at 70% amplitude in 0.5 s on, 0.5 s off cycles for 10 min. Cell debris was removed by centrifugation at 20,000 × g and 4 °C for 20 min. SiaP was purified at 4 °C by anion-exchange chromatography using a Q 16/10 anion exchange column (Cytiva) equilibrated in 50 mM Tris-HCl pH 8.0. The column was washed with 5 CV equilibration buffer and bound protein was eluted with a gradient to 50 mM Tris-HCl pH 8.0, 1 M NaCl across 10 CV. Ammonium sulfate was added to the eluted SiaP to a final concentration of 1 M. SiaP was further purified by hydrophobic-interaction chromatography and bound to a Phenyl FF 16/10 column (Cytiva) equilibrated in 50 mM Tris-HCl pH 8.0 and 1 M ammonium sulfate. The column was washed with 5 CV equilibration buffer and bound protein was eluted with a gradient to 50 mM Tris-HCl pH 8.0, 150 mM NaCl across 10 CV. Eluted protein was concentrated for SEC using 10 kDa MWCO spin concentrators (Pall) and loaded onto a HiLoad 16/60 Superdex 200 size-exclusion column (Cytiva) equilibrated with 50 mM Tris-HCl pH 8.0, 150 mM NaCl and 1 mM N-acetylneuraminic acid (Carbosynth). For ITC and thermal shift experiments, SEC was performed without N-acetylneur-aminic acid. SEC fractions containing SiaP were pooled and con-centrated. Protein was either used in subsequent experiments, or flash-frozen in liquid nitrogen and stored at −80 °C for future use.

## Reconstitution of SiaQM from *Photobacterium profundum* in proteoliposomes

Purified SiaQM was reconstituted using a batch-wise detergent removal procedure as previously described[33,34]. In brief, 50 µg of SiaQM was mixed with 120 µL 10% C$_{12}$E$_8$, 100 µL of 10% egg yolk phospholipids (w/v), in the form of sonicated liposomes as previously described[46] (except where differently indicated in the figure legend), 50 mM of K$^+$-gluconate, 20 mM HEPES/Tris pH 7.0 in a final volume of 700 µL. The reconstitution mixture was incubated with 0.5 g Amberlite XAD-4 resin under rotatory stirring (1200 rev/min) at 25 °C for 40 min[46].

## Transport measurements and transport assay

After reconstitution, 600 µL of proteoliposomes were loaded onto a Sephadex G-75 column (0.7 cm diameter × 15 cm height) pre-equilibrated with 20 mM HEPES/Tris pH 7.0 with 100 mM sucrose to balance the internal osmolarity. Then, valinomycin (0.75 µg/mg phospholipid) prepared in ethanol was added to the eluted proteoli-posomes to generate a K$^+$ diffusion potential, as previously

described[33]. After 10 s of incubation with valinomycin, transport was started by adding 5 µM [$^3$H]-Neu5Ac to 100 µL proteoliposomes in the presence of 50 mM Na$^+$-gluconate and 0.5 µM of SiaP. For kinetic measurement, the initial transport rate was measured by stopping the reaction after 15 min (or as stated in the figure legends), i.e., within the initial linear range of [$^3$H]-Neu5Ac uptake into the proteoliposomes as determined by time course experiments. The transport assay was terminated by loading each proteoliposome sample (100 µL) on a Sephadex G-75 column (0.6 cm diameter × 8 cm height) to remove the external radioactivity. Proteoliposomes were eluted with 1 mL 50 mM NaCl and collected in 4 mL of scintillation mixture, vortexed and counted. The radioactivity taken up in controls performed with empty liposomes, i.e., liposomes without incorporated protein, was negligible with respect to the data obtained with proteoliposomes, i.e., lipo-somes with incorporated proteins. Data analysis was performed by *Grafit* software (version 5.0.13) using the Hill plot for kinetics deter-mination and the first-rate order equation for time course analysis. When measuring the dependence of Neu5Ac uptake into proteolipo-somes with SiaP, the transport was measured in the presence of varying concentrations of SiaP and measured after 30 min. All mea-surements are presented as means ± s.e.m. from independent experi-ments as specified in the figure legend. Graphs of data and fit were produced using *GraphPad Prism* (version 9).

## Crystallography, small angle X-ray scattering data collection and analysis

Purified SiaP in SEC buffer was concentrated to 50 mg mL$^{-1}$ using a 10 kDa MWCO spin concentrator and crystallised using the sitting-drop vapour-diffusion method at 20 °C. The best diffracting crystals were grown in the E1 condition from the Shotgun screen (2.0 M ammonium sulfate, 0.1 M Bis-Tris pH 5.0) (Molecular Dimensions). Crystals were cryo-protected by soaking in reservoir solution supple-mented with liquid glycerol and ethylene glycol.

Crystallographic data were collected on the MX2 beamline at the Australian Synchrotron at a wavelength of 0.95372 Å, using an Eiger 16M detector (ACRF ANSTO). Data were scaled and processed using *XDS*[47] and *CCP4*[48]. Phases were obtained by molecular repla-cement using *Phaser*[49] with 2xwk as the search model. Model building was performed using *coot*[50], and refinement performed with *phenix*[51]. Atomic displacement parameters (ADPs) were refined anisotropically. *PDB-redo* was used to optimise the model. The final refined model had 98.49% favoured and zero outliers in the Rama-chandran plot. The structure was deposited into the PDB with the identification code 7t3e.

Small angle X-ray scattering data were collected on the SAXS/WAXS beamline equipped with a Pilatus 1 M detector (170 mm × 170 mm, effective pixel size, 172 µm × 172 µm) at the Australian Syn-chrotron. A sample detector distance of 1600 mm was used, pro-viding a q range of 0.05–0.5 Å$^{-1}$. Here 70 µL of purified SiaP protein at 10 mg mL$^{-1}$ was injected onto an inline Superdex S200 Increase 5/150 GL (GE Healthcare) SEC column, equilibrated with 20 mM Tris-HCl pH 8.0, 150 mM NaCl, 0.1% (w/v) NaN$_3$, with or without 10 mM Neu5Ac using a flow rate of 0.45 mL min$^{-1}$. Scattering data were col-lected in one sec exposures (λ = 1.0332 Å) over a total of 400 frames, using a 1.5 mm glass capillary, at 8 °C. 2D intensity plots were radially averaged, normalised to sample transmission, and background sub-tracted using the *Scatterbrain* software package (Australian Synchrotron).

## Isothermal titration calorimetry

Isothermal titration calorimetry (ITC) experiments were per-formed using a TA Instruments Low Volume NanoITC. Protein and ligand samples were prepared in buffer comprised of 50 mM Tris-HCl pH 8.0, and 150 mM NaCl. Protein concentrations were mea-sured using a Thermo Scientific™ NanoDrop™ One Microvolume

UV-Vis Spectrophotometer. The sample cell was loaded with 90 μM SiaP and the injection syringe was loaded with 600–750 μM Neu5Ac. After the system was equilibrated at 25 °C with a stirring speed of 400 rpm, titrations were started with a single injection of 1 μL, followed by 2 μL injections every 180 s. To determine the thermodynamic values, global fit analysis was carried out using *SEDPHAT*[52], by fitting the curves of technical triplicate experiments into a single site binding isotherm, with the first injection excluded. The graph of data and fit was produced using *GraphPad Prism* (version 9).

## Nanobody screening and megabody design
Nanobodies were selected using methods modified from McMahon et al.[21]. In short, SiaQM protein was purified and modified with amine reactive FITC- and Alexafluor647-labels (A647; ThermoScientific). Labelling was verified using spectrophotometry and equated to approximately 1.8 and 2.0 fluorescent label per protein molecule, respectively. Nanobodies were selected in buffer comprised of 20 mM HEPES pH 7.6, 150 mM sodium chloride, 0.1% (w/v) bovine serum albumin, 5 mM maltose, 0.002% L-MNG and 2 mM Neu5Ac. To negatively select non-specific clones, the naive nanobody library was incubated with 400 μL anti-A647 magnetic beads (Miltenyi Biotec), immobilised in an LD column, and the flowthrough incubated with 1 μM A647-labelled SiaQM. SiaQM-binders were enriched by adding 400 μL of anti-A647 magnetic beads (or anti-FITC beads as appropriate), and captured in LS columns (Miltenyi Biotec), then released into Yglc4.5-Trp media for recovery. After one day of recovery, yeast were transferred to -Trp glucose media and after 24 h nanobody expression was induced with galactose for 24 h prior to selection. The selection procedure encompassed two passes of three selection types −successive rounds of magnetic selection with anti-A647, then anti-FITC protein capture, and a third selection by fluorescence assisted cell sorting (FACS) using A647-labelled protein. Selection was progressively more stringent over these rounds, with protein concentrations of, 1 μM, 500, 200, 60, 30, 10 nM. After the final round of FACS, a dilution series of the recovered cells were plated on YPD agar. Single colonies were grown in 96-well plates and amplified for Sanger sequencing using a modified primer pair (for-GAAGGTGTTCAATTG-GACAAGAGAGAAGCTGAC; rev-GCGTAATCTGGAACATCGTATGGGTA GGATCC). Sequences were aligned using *Geneious*[53]. Nanobodies were selected for follow-up based on enrichment at the sequence level and confirmation of labelled SiaQM binding to individual yeast clones assessed by flow cytometry. The successful sequence of Nb07 comprised 12 of the 96 (12.5%) sequenced clones from the final sequenced plate.

The megabody (Mb$_{Nb07}^{HopQ}$) was constructed using a reported protocol[20], as follows: residues 1–13 of the nanobody β-strand A (residues 1–13), followed by the C-terminal domain of *Helicobacter pylori* HopQ (Uniprot ID: B5Z8H1) (residues 227–446), which was directly fused to the N-terminal domain of HopQ (residues 53–221) followed by the remainder of the nanobody (residues 14–122) (Supplementary Table 7). The sequence encoding the megabody Mb$_{Nb07}^{c7HopQ}$ was synthesised by Genscript, and cloned into the expression vector pET-22B(+), encoding an N-terminal *pelB* signal peptide for periplasmic targeting and a C-terminal hexahistidine tag. Protein overexpression and purification were carried out using established methods[20].

## Amphipol exchange
SEC-purified SiaQM was incubated with amphipol A8-35 (Anatrace) at a 1:5 w/w ratio for 2 h before addition of 100 mg mL$^{-1}$ Bio-Beads SM-2 resin (Bio-Rad) and then incubated overnight at 4 °C with gentle agitation to remove detergent. After exchange, SEC was performed in 50 mM Tris-HCl pH 8.0 and 150 mM NaCl buffer to remove free amphipol and assess protein monodispersity.

## Nanodisc reconstitution
MSP1D1 with a hexahistidine tag and TEV cleavage site was purified as previously described[54]. Powdered lipid extract was resuspended in 50 mM Tris-HCl pH 8.0, 150 mM NaCl to a final concentration of 20 mM. For the reconstitution, SEC-purified SiaQM was incubated with MSP1D1 and *E. coli* polar lipid extract (Avanti) at a ratio of 1:5:100, respectively. The mixture was incubated together for 30 min at room temperature, then transferred to a tube containing 50 mg of Bio-Beads SM-2 resin (Bio-Rad) and incubated for a further 2 h at room temperature, followed by incubation overnight at 4 °C using a rotary shaker. To remove empty nanodiscs, the solution was incubated with Ni-NTA resin (approximately 1 mL per mg of SiaQM) for 1 h at 4 °C. The resin was then washed with five CVs of 50 mM Tris-HCl pH 8.0, 150 mM NaCl. Bound nanodisc-reconstituted SiaQM was eluted in two CVs of 50 mM Tris-HCl pH 8.0, 150 mM NaCl, 250 mM imidazole buffer and then subjected to SEC in 50 mM Tris-HCl pH 8.0, 150 mM NaCl.

## Megabody complex formation
SiaQM in amphipol and nanodiscs were complexed with the megabody Mb$_{Nb07}^{c7HopQ}$ at a 1:1.5 molar ratio. SEC using a 24 mL Superdex 200 increase 10/300 GL was used to further purify the SiaQM-Mb$_{Nb07}^{c7HopQ}$ complex, in 50 mM Tris-HCl pH 8.0, 150 mM NaCl. Fractions containing the SiaQM−Mb$_{Nb07}^{c7HopQ}$ complex (Supplementary Fig. 10) were concentrated, flash-frozen in liquid nitrogen and stored at −80 °C for cryo-EM experiments.

## Assessment of complex formation
Sedimentation velocity analytical ultracentrifugation was used to assess complex formation with three samples. SiaQM (3.5 μM) in amphipol, 5.25 μM Mb$_{Nb07}^{c7HopQ}$, and both combined (1:1.5 molar ratio). The size-exclusion buffer was 50 mM Tris-HCl pH 8.0, 150 mM NaCl.

## Single-particle cryo-EM vitrification and data acquisition
SiaQM-Mb$_{Nb07}^{c7HopQ}$ in the presence of 10 mM Neu5Ac was concentrated to 2.9 mg mL$^{-1}$ (amphipol) or 1 mg mL$^{-1}$ (nanodiscs) and 3 μL of sample was applied to freshly glow-discharged (Gatan Solarus) Quantifoil R2/1 cu300 mesh grids (Electron-microscopy Sciences). Grids were blotted using a Vitrobot Mark IV (Thermofisher Scientific) for 3.5 s, at 4 °C with 100% humidity before vitrification in liquid ethane. Cryo-EM datasets were collected on a Titan Krios G3i electron microscope equipped with a K3 detector and BioQuantum imaging filter (Gatan), operated at 300 kV in counting mode. Movie stacks were collected at a nominal magnification of ×130,000 and a 0.6645 Å pixel size, with a dose rate of 16.57 e⁻/pixel/s (amphipol) or 15.64 e⁻/pixel/s (nanodisc). Each movie was a result of 1.9 s exposures with a total accumulated dose of 71.3 e⁻/Å² (amphipol) or 2 s exposures with a total accumulated dose of 70.9 e⁻/Å² (nanodisc), which were fractionated into 40 frames. The *EPU* software package (Thermo Fisher Scientific) was used for automated data collection and an energy filter slit width of 20 eV and a 50 μm C2 condenser aperture were used during imaging, while the objective aperture was not inserted. In total, 10,960 micrographs were recorded for the dataset in amphipol and 8127 micrographs were recorded for the dataset in nanodisc. The statistics for cryo-EM data acquisition are summarised in Supplementary Table 1.

## Single-particle cryo-EM data processing and map refinement
Unless otherwise stated, all cryo-EM data processing was performed using *CryoSPARC* v.3.2.0[55] (Supplementary Fig. 2). For both datasets, movie frames were aligned using patch motion correction with a *B*-factor of 500, then contrast transfer function (CTF) estimations were made using the patch CTF estimation tool. Particles were first picked from 300 micrographs using the Blob Picker tool and extracted. These particles were 2D classified into 50 classes and the best 2D classes selected and used as a template for automated particle picking using the Template Picker tool. Particles were inspected with the Inspect

Picks tool and a total of 5,609,037 particles were extracted with a box size of 300 pixels Fourier cropped to 200 pixels for the amphipol dataset, and 2,922,552 particles were extracted with a box size of 400 pixels Fourier cropped to 200 pixels for the nanodisc dataset. For each dataset, the extracted particles were then sorted using iterative rounds of 2D classification and the best 2D classes showing some structural details were selected. The selected particles were subjected to ab initio reconstruction separated into multiple classes. The best 3D reconstruction for the dataset in amphipol contained 624,033 particles and was used as a reference model for an initial round of non-uniform refinement, allowing a 3.29 Å model to be reconstructed. The best 3D reconstruction for the dataset in nanodiscs contained 499,085 particles and was used as a reference model for an initial round of non-uniform refinement, allowing a 3.13 Å model to be reconstructed. Particles were re-extracted with a final box size of 300 pixels Fourier cropped to 240 pixels, giving a final pixel size of 0.83 Å for the dataset in amphipol. Particles were re-extracted with a final box size of 400 pixels and Fourier cropped to 300 pixels, giving a final pixel size of 0.89 Å for the dataset in nanodisc. Iterative rounds of local refinement followed for both datasets, using a mask that included SiaQM and nanobody, and excluding the surrounding amphipol, using an initial low pass resolution of 10 Å, searching over a range of 1° in orientations and 1 Å shifts. For the dataset in nanodisc, the mask included both SiaQM and the surrounding belt protein.

### Structural model building and analysis

The atomic model of SiaQM-Mb$_{Nb07}^{c7HopQ}$ was built de novo from the globally sharpened 2.97 Å map using the *phenix* Map to Model tool[56]. Segments were manually joined using *coot* (version 0.9.5 EL) and the structure was refined using *phenix* real space refine using secondary structure, rotamer and Ramachandran restraints. The *Namdinator* tool[57] was used between rounds of manual model building to optimise geometry and reduce clashes. Models were validated using Molprobity within *phenix*[45]. The structure was deposited into the PDB with identification code 7qha and the EMDB with the identification code EMD-13968. For the structure embedded within a nanodisc, the structure was deposited into the PDB with identification code 8b01 and the EMDB with the identification code EMD-15775.

The SiaQM:SiaP complex was predicted using *AlphaFold*[37] (using all three sequences as inputs, with no fusion sequences), with experimentally determined structures then aligned to this complex. *Rosetta fast relax*[58,59] was then used to improve the sidechain packing of the complex. This model can be found in Supplementary Data 1.

The outward-facing model of SiaQM was generated with *Modeller*[60] using PDB ID: 6wu4 as a template for the transport domain. This model can be found in Supplementary Data 2.

### Analytical ultracentrifugation

Sedimentation velocity experiments were performed using an XL-I analytical ultracentrifuge (Beckman Coulter). Reference solution (400 μL) and sample solutions (380 or 400 μL) were loaded into cells with double sector 12 mm Epon centrepieces and sapphire windows. The samples were run in an An-60 Ti rotor at 42,000 rpm and 20 °C until all species had completely sedimented. Both absorbance (280 nm) and interference optical systems were used to monitor sedimentation. Data analysis was performed with *Ultrascan 3* (version 4.0)[61], *Sedfit*[62] and *GUSSI*[63].

### Thermal shift assay

The melting temperature of wildtype and mutant protein samples (1 mg mL$^{-1}$) were measured in 50 mM Tris-HCl pH 8.0 and 150 mM NaCl in the absence of Neu5Ac and with 1 mM Neu5Ac. Protein unfolding was monitored using 10X SYPRO Orange dye (Invitrogen) with a QuantStudio 3 Real-Time PCR System (Applied Biosystems). Each sample was measured in triplicate across a 20–80 °C range with

a heating rate of 0.03 °C/s. Data were analysed using the derivative function with *Protein Thermal Shift Software* v1.4 (Applied Biosystems).

### Spectrofluorometric assays

The intraliposomal sodium ion accumulation was monitored by measuring the fluorescence emission of Sodium Green™. After reconstitution, 600 μL of proteoliposomes was passed through a Sephadex G-75 column (0.7 cm diameter × 15 cm height) pre-equilibrated with 20 mM HEPES/Tris pH 7.0. After elution from Sephadex G-75 column, valinomycin (0.75 μg/mg phospholipid) prepared in ethanol was added to the proteoliposomes to impose the membrane potential. Then, uptake experiments were started in a 150 μl proteoliposome sample by adding 5 μM Neu5Ac together with 75 mM Na-gluconate and 0.5 μM SiaP, at 25 °C. The transport reaction was stopped at 20 min; to separate the external from the internally accumulated Na$^+$, samples were passed through a Sephadex G-75 column (0.6 cm diameter × 8 cm height) buffered with 20 mM HEPES/Tris pH 7.0, 20 mM sucrose; samples were then, eluted with 1 mL of the same buffer. The fluorometric assay was performed in 2 mL of eluted proteoliposome from the previous step, with a mixture of 50 nM Sodium Green™ indicator (SGI) and 0.25% C$_{12}$E$_8$ to destabilise proteoliposomes, allowing SGI to interact with internally accumulated Na$^+$. The measurements were performed in the fluorescence spectrometer (LS55) from Perkin Elmer with stirring. The fluorescence was measured following the time drive acquisition protocol with λ excitation = 507 nm and λ emission = 532 nm (slit 5/4) according to manufacturer instructions of Sodium Green™. The described end-point strategy has been used in place of a real-time approach due to the chemical complexity of the SGI that unspecifically binds to liposomes. Calibration of the fluorescence changes versus released Na$^+$ has been performed by measuring the fluorescence of known amounts of Na$^+$-gluconate (from 0 to 2000 nmol in 2 mL) obtaining a linear correlation as previously performed[28,64]. The calibration curve was used to calculate the nmoles of Na$^+$ taken up in proteoliposomes reconstituted with SiaQM from *P. profundum*. A calibration was performed at the end of each experiment.

### Reporting summary

Further information on research design is available in the Nature Portfolio Reporting Summary linked to this article.

## Data availability

The crystal structures and data are available from the PDB under the code 7T3E and the cryo-EM structures and data are available from the PDB under the codes 7QHA and 8B01, and the EMDB under the codes EMD-13968 and EMD-15775). Source data are provided with this paper.

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

## Acknowledgements

This research was undertaken in part using the MX2 beamline at the Australian Synchrotron, part of the Australian Nuclear Science and Technology Organisation (ANSTO), and made use of the Australian Cancer Research Foundation (ACRF) detector as well as the SAXS beamline at the Australian Synchrotron, part of ANSTO. R.C.J.D., R.A.N., J.R.A., S.W., J.S.D. and M.J.C. acknowledge funding support from the Marsden Fund, managed by Royal Society Te Apārangi (contract UOC1506) and the Biomolecular Interaction Centre (UC). R.C.J.D., J.S.D. and M.J.C. also acknowledge the following for funding support, in part: (1) the Ministry of Business, Innovation and Employment Smart Ideas grant (contract UOCX1706); (2) the Maurice Wilkins Centre flexible research grant; and (3) the Australian Institute of Nuclear Science and Engineering (AINSE Ltd) and ANSTO for a Postgraduate Research Award. R.A.N. acknowledges the Canterbury Medical Research Fund (contract CMRF 08). J.R.A. acknowledges the Rutherford Discovery Fellowship, managed by the Royal Society Te Apārangi (contract 15-MAU-001/15-UOA-008). J.C. is supported by a University of Auckland Doctoral Scholarship. The authors acknowledge the facilities, and scientific and technical assistance from flow cytometry staff at Otago Micro and Nanoscale Imaging (OMNI), at the University of Otago. We thank Prof. Borries Demeler (University of Lethbridge, Canada) for help with AUC experiments. R.F. acknowledges the Swedish Governmental Agency for Innovation Systems (2017-00180), and the Centre for Antibiotic Resistance Research (CARe) at the University of Gothenburg. C.I. acknowledges the MIUR (Ministry of Education, University and Research) Italy for the support through the "SI.F.I.PA.CRO.DE. – Sviluppo e industrializzazione farmaci innovativi per terapia molecolare personalizzata PA.CRO.DE." (PON ARS01_00568). D.D. acknowledges funding from the Knut and Alice Wallenberg Foundation. Cryo-EM data were collected at the Cryo-EM Swedish National Facility funded by the Knut and Alice Wallenberg Foundation, the Family Erling Persson and Kempe Foundations, SciLifeLab, Stockholm University and Umeå University. We thank Aashish Manglik (University of California) and Andrew Kruse (Harvard University) for providing the original nanobody yeast-display library.

## Author contributions

Project designed by R.A.N. and R.C.J.D. Research funding was obtained by R.C.J.D., R.A.N., P.D.M., J.R.A. and S.W. Protein expression and purification were performed by J.S.D., M.J.C., R.A.N., M.C.N.-V., J.W., D.M.R. and G.A. Grid optimisation was carried out by R.A.N. and Cryo-EM data collection was performed by R.A.N., J.S.D. and D.R.M. Cryo-EM data processing, map refinement and 3D reconstruction were performed by R.A.N., J.S.D. and A.G. Model building and structure building was carried out by J.S.D., with refinement, analysis and interpretation by J.S.D. and M.J.C. Nanobody screening and selection was performed by S.A.J. and P.D.M. Nanobody production and megabody design for Cryo-EM was carried out by J.S.D. AUC experiments were conducted and analysed by M.J.C. ITC experiments were conducted and analysed by M.C.N.-V. Proteoliposome assays were performed and analysed by M.S. and C.I. X-ray crystallography experiments, data processing, and structure determination were performed by J.S.D. and R.A.N. Refinement was performed by J.S.D. and J.W. Bioinformatic analysis was performed by J.W. and J.S.D. Structural modelling, docking and interpretation of the complex was performed by J.S.D. and J.C. Overall experimental development, analysis of data and interpretation of the results was overseen by R.A.N., S.R., R.F., S.W., J.R.A., C.I., D.D., P.D.M. and R.C.J.D. The manuscript was prepared by J.S.D., M.J.C., R.A.N., D.D. and R.C.J.D. with contributions from all authors.

## Competing interests

R.F. is currently employed by AstraZeneca.

## Additional information

[1]Biomolecular Interaction Centre, Maurice Wilkins Centre for Biodiscovery, MacDiarmid Institute for Advanced Materials and Nanotechnology and School of Biological Sciences, University of Canterbury, PO Box 4800 Christchurch 8140, New Zealand. [2]Department of Biochemistry and Biophysics, Stockholm University, 10691 Stockholm, Sweden. [3]Department DiBEST (Biologia, Ecologia, Scienze della Terra) Unit of Biochemistry and Molecular Biotechnology, University of Calabria, Via P. Bucci 4C, 87036 Arcavacata di Rende, Italy. [4]Biomolecular Interaction Centre, Digital Life Institute, Maurice Wilkins Centre for Molecular Biodiscovery, and School of Biological Sciences, University of Auckland, Auckland 1010, New Zealand. [5]Science for Life Laboratory, Department of Biochemistry and Biophysics, Stockholm University, 17165 Solna, Sweden. [6]Biochemistry Department, School of Biomedical Sciences, University of Otago, Dunedin 9054, New Zealand. [7]Biological Sciences and Biomedical Engineering, Bindley Bioscience Center, Purdue University, 1203 W State St, West Lafayette IN 47906, USA. [8]Centre for Antibiotic Resistance Research (CARe) at University of Gothenburg, Box 440, S-40530 Gothenburg, Sweden. [9]Biological Sciences Division, SLAC National Accelerator Laboratory, Menlo Park, CA 94025, USA. [10]Department of Structural Biology, Stanford University School of Medicine, Stanford, CA 94305, USA. [11]CNR Institute of Biomembranes, Bioenergetics and Molecular Biotechnologies (IBIOM), Via Amendola 122/O, 70126 Bari, Italy. [12]Bio21 Molecular Science and Biotechnology Institute, Department of Biochemistry and Molecular Biology, University of Melbourne, Parkville, Victoria 3010, Australia. [13]The authors contributed equally: James S. Davies, Michael J. Currie, Rachel A. North. ✉e-mail: rachel.north@dbb.su.se; renwick.dobson@canterbury.ac.nz

