## [Peer Review File · Nature Communications]

Structure and mechanism of a tripartite ATP-independent periplasmic TRAP transporterEditorial Note: This manuscript has been previously reviewed at another journal that is not operating a transparent peer review scheme. This document only contains reviewer comments and rebuttal letters for versions considered at *Nature Communications*.

REVIEWERS' COMMENTS

Reviewer #1 (Remarks to the Author):

The revised manuscript has significantly improved with the additional functional data to support their claims.

The structure and functional characterisation of the H. influenzae SiaQM transporter was published (<https://doi.org/10.1038/s41467-022-31907-y>) before this revision was submitted. The authors have made no effort to discuss the HiSiaQM in the context of their work. The HiSiaQM provides complementary insights to this manuscript. A substantial discussion between the two systems is necessary.

Reviewer #2 (Remarks to the Author):

The authors have improved the manuscript substantially in response to the comments of the reviewers. In my opinion it is now publishable. One thing the authors could still change is the representation of the QM transporter in the scheme of Fig. 4. Currently, the diagrams are confusing because it looks like there are three different proteins in the membrane.

Otherwise, this is a nice paper that extends our understanding of an interesting family of secondary active transporters.

Reviewer #3 (Remarks to the Author):

Comments on “Structure and mechanism of the tripartite ATP-independent periplasmic (TRAP) transporter” by Davies et al.

Major comments:

Page 3 line 7: “how TRAP transporters work is poorly understood” – This is not entirely true/ a too simple statement which excludes many studies. Since the submitted manuscript can be seen as a milestone in the field of TRAP transporter, the introduction should be more detailed to introduce the current knowledge about TRAP transporters to the readers (Further/other literature comments from the introduction can be find below). In addition, a recently published structure of another TRAP transporter is not mentioned, although the authors cite this work in ref. 35 as “functional study” on TRAP transporters (so the work should be familiar with this publication). Since ref 35 is more than just a functional study, much more effort should be made to compare the current work with ref 35. In addition, I would recommend to give a better overview about the existing TRAP transporter studies:

Page 2 line 29: the listed references 7-10 are navigating all to SiaP/substrate binding studies. It would be more appropriate to cite (or additional cite) the following publications, which include some gene analysis or focused on SiaQM interactions:

1. Kelly and Thomas, 2001, FEMS Microbiology Reviews 20 (2001) 405-424.
2. Mulligan et al., 2007, J Mol Microbiol Biotech, 12, 218-226
3. Mulligan et al., 2012, JBC, 287, 5, 3598
4. Rosa et al., 2018, frontiers in cellular and infection microbiology, 8, 33.

Page 2 line 30: I don't understand why the authors are citing here a classification of binding proteins. Would be more appropriate to cite studies on the transport mechanism of TRAP transporters which showed the dependence of the P-subunit:

1. First in vitro liposome assay: Mulligan et al., 2009, PNAS, 106, 6.
2. Complementation analysis: Forward et al., 1997, J Bacter, 179, 17.
3. Substrate uptake assay: Severi et al., 2005, Mol Microbiol, 58,4,1173-1185.
4. Bactericidal assay: Johnston et al., 2008, JBC, 283,2,855.

Page 6 line 28...: The interpretation of the data on the interaction between SiaP and SiaQM remains inconsistent. MTS shows a K_d of 400 μM in amphipol, but AUC shows no interaction. The obvious explanation would be that AUC is not sufficiently sensitive to detect such weak interaction. Yet, in detergent solution a K_d of 400 μM is detectable using AUC. Because an interaction with K_d of 400 μM is “transient” regardless of the use of amphipol or detergent, the discrepancy is not properly addressed. My two cents' worth: the MTS data cannot be used to determine a reliable K_d (because there is no

indication on the plateau level), so it may be may higher than 400 μM . The problem is exacerbated because: 1. The fits of the MTS data have huge errors, and only duplicate experiment have been done, where not even error propagation was attempted; 2. MTS was not done for the detergent sample, which makes comparison between the techniques impossible; 3. The AUC measurements, at least in panel f for L-MNG where an interaction was observed, are presented as single experiments. Especially since the signal is very small, the validity of the interpretation can be questioned without proper replicate experiments.

Additional issues regarding the tripartite complex analysis data (concerning Extended Data Figure 1):

- Additionally, the authors should also perform the MST measurements as triplicates, especially due to the large variety of measured data points (in contrast, the high-quality ITC experiments in Ex Data Fig 8 are done as triplicates!).
- Laser interferometry was used to determine the detergent concentration, however no experimental data is shown, nor the experiment is mentioned in the method part.
- FITC-labelled SiaP was used but the labelling is not mentioned in the method part

Page 5 line 20: The two-fold increased activity is very interesting – however the authors should consider that this effect might be also caused by different reconstitution efficiencies of SiaQM. The authors should provide a reconstitution efficiency or at least mention this as possible reason for the detected difference in the main text. In addition, the liposome assays are all performed in the lower activity egg-PC. Why? If the authors figured out that the protein is significantly more active in *E. coli* lipids, it might be more appropriate to perform the experiments in these lipids.

Page 7 line 17: It is known since Severi et al. 2005, that SiaP is contaminated with sialic acid if the protein is expressed in LB medium (due to the casein). Protein for apo structure determination (PDB 2ECY) or analysis of the conformational transition (Glaenger et al., 2017, Biophysical Journal; Peter et al., 2021, JMB) are therefore mostly expressed in minimal medium. It is not clear if the protein prepared for ITC experiments came from minimal medium expression. In addition, the authors explain the purification of SiaP in the Method chapter with addition of 1 mM sialic acid in the SEC run, what is not compatible with subsequent substrate binding studies (the new ITC studies and the thermal shift assay experiments).

Title: I would recommend changing it to a more general description, since there is not THE Trap transporter analysed in this study (maybe use "... description of a TRAP transporter")

Page 2 line 3-5 and line 30: This is not correct, there is another class of secondary active transporters, TTTs, which share no sequence identity to TRAP transporters but also have a substrate binding protein.

Page 4 line 34 and Page 5 line 1-3: I agree that the Q-subunit seems to extend the scaffold. However, the next sentence sounds like there are correlations between the lipid density and the Q-subunit scaffold properties. This can easily confuse readers, especially since there are also lipids next to the static transport domain (page 5 line 24).

Page 5 line 21: The E. coli lipid reconstitution is not explained in the corresponding method chapter (just with egg yolk phospholipids page 25 line 11).

Page 5 line 26: For the whole manuscript, “sodium” is often used incorrectly to refer to “sodium ions”

Page 7 line 9-10: SiaP, SiaM and SiaQ are three separate peptide chains. The authors should give more details (at least in the method chapter) how they used for example AlphaFold to predict the complex. Were the peptides fused into one chain?

Page 7 line 23: The thermal shift assay from extended data figure 11 should be already mentioned here for mutant R49A and E170A (I think there is a typo in E170A or should this mutant firstly mentioned in line 2 page 9?).

Page 9 line 19 and figure 3e: It is very hard to see any upward movement in the transport domain in this representation.

Figure 1: B: sigma level is missing

Page 24 line 22: Give the temperature for the 2h solubilization step in L-MNG. Also check the purification description of SiaQM and SiaP and add temperatures to all steps, for example lysis, affinity chromatography and SECs.

Ext Data Fig 8e: The ITC measurement looks very nice, but the authors should also include the ΔH and ΔS from the triplicate measurements (only the confidence interval of the KD is mentioned in the figure description).

We thank the reviewers for their constructive comments, which we have incorporated into our newly revised manuscript entitled “**Structure and mechanism of a tripartite ATP-independent periplasmic TRAP transporter**” by Davies, Currie *et al.*

We have addressed each reviewer comment (shown in *black italics*) in our point-by-point response below, where we respond in **blue text**, with changes underlined in this letter. We have supplied a revised version of the manuscript including these changes (with track changes).

Reviewer #1:

“The revised manuscript has significantly improved with the additional functional data to support their claims.”

Minor point:

1.1 *“The structure and functional characterisation of the *H. influenzae* SiaQM transporter was published <https://doi.org/10.1038/s41467-022-31907-y>) before this revision was submitted. The authors have made no effort to discuss the HiSiaQM in the context of their work. The HiSiaQM provides complementary insights to this manuscript. A substantial discussion between the two systems is necessary.”*

Response. We have strengthened our discussion of this publication, focusing on the accompanying functional work. We find that the majority of the *in vivo* and mutagenesis work performed by Peter *et al* supports our transport assay and mechanistic interpretations and have included a thorough discussion of this. We have chosen to limit discussion of the reported structure because of the low resolution, which does not allow unambiguous modelling of sidechains and loops, as shown below.

Here we compare the high quality of our data (yellow and blue). In grey, we display data recently published in *Nat. Commun.* Although presented as the ‘first structure’ of a TRAP transporter, these data do not allow unambiguous modelling of the protein chain, and rely heavily on an AlphaFold model to ‘fill in the gaps’.

We have, however, included a thorough discussion of the functional data. See the following Page 8 lines 3-5. “This finding is supported by recent work by Peter *et al.*, who show that mutation of the equivalent residues in HiSiaPQM (R484 and S60) affects growth in an *E. coli* complementation experiment³⁹.”

Page 8 lines 31-33. “Analysis of this region by Peter *et al.* only showed a minor growth defect with N150D in an *in vivo* assay, perhaps suggesting this mutation is not strong enough to fully lock the SiaP in a closed conformation³⁹.”

Page 9 lines 2-4. “This result is supported by a charge swap mutation of the equivalent residue in *HiSiaP* (E172) to an arginine, which gives a significant growth defect³⁹.”

Page 9 lines 30-34. “This motion and mechanism are consistent with that proposed by Peter *et al.*³⁹, who used a similar approach utilising a model of the *VcINDY* outward state. Here the authors also present a 4.7 Å cryo-EM structure of the fused *HiSiaQM* system. Our structures confirm the reported fold (see **Extended Data Figure 12** for an overlay), and provide further detail at the residue level, allowing us to identify ion and lipid binding sites and resolve the loop regions.”

Reviewer #2:

“The authors have improved the manuscript substantially in response to the comments of the reviewers. In my opinion it is now publishable. ... Otherwise, this is a nice paper that extends our understanding of an interesting family of secondary active transporters.”

Minor point:

2.1 *“One thing the authors could still change is the representation of the QM transporter in the scheme of Fig. 4. Currently, the diagrams are confusing because it looks like there are three different proteins in the membrane.”*

Response. We thank the reviewer for noticing this, and have updated the Figure to show clearly that there are only two protein chains in the membrane.

Reviewer #3:

3.1 *“Page 3 line 7: “how TRAP transporters work is poorly understood” – This is not entirely true/ a too simple statement which excludes many studies. Since the submitted manuscript can be seen as a milestone in the field of TRAP transporter, the introduction should be more detailed to introduce the current knowledge about TRAP transporters to the readers (Further/other literature comments from the introduction can be find below).”*

Response. We acknowledge that this statement does not fully address foundational studies on TRAP transporters, which we have now included in our references. We have modified this statement to “not fully understood”.

3.2 *“In addition, a recently published structure of another TRAP transporter is not mentioned, although the authors cite this work in ref. 35 as “functional study” on TRAP transporters (so the work should be familiar with this publication). Since ref 35 is more than just a functional study, much more effort should be made to compare the current work with ref 35. In addition, I would recommend to give a better overview about the existing TRAP transporter studies:”*

Response. We agree that this statement does not acknowledge foundational studies on TRAP transporters, which we have now included in our references. We have modified this statement to “not fully understood”. We have made more effort to compare the current work with ref 35, as per Reviewer 1’s suggestion.

3.3 *“Page 2 line 29: the listed references 7-10 are navigating all to SiaP/substrate binding studies. It would be more appropriate to cite (or additional cite) the following publications, which include some gene analysis or focused on SiaQM interactions:*

1. Kelly and Thomas, 2001, *FEMS Microbiology Reviews* 20 (2001) 405-424.
2. Mulligan *et al.*, 2007, *J Mol Microbiol Biotech*, 12, 218-226
3. Mulligan *et al.*, 2012, *JBC*, 287, 5, 3598
4. Rosa *et al.*, 2018, *frontiers in cellular and infection microbiology*, 8, 33.

Response. These have been included.

3.4 *“Page 2 line 30: I don’t understand why the authors are citing here a classification of binding proteins. Would be more appropriate to cite studies on the transport mechanism of TRAP transporters which showed the dependence of the P-subunit:*

1. First in vitro liposome assay: Mulligan et al., 2009, PNAS, 106, 6.
2. Complementation analysis: Forward et al., 1997, J Bacter, 179, 17.
3. Substrate uptake assay: Severi et al., 2005, Mol Microbiol, 58,4,1173-1185.
4. Bactericidal assay: Johnston et al., 2008, JBC, 283,2,855.”

Response. These have been included.

3.5 “Page 6 line 28...: The interpretation of the data on the interaction between SiaP and SiaQM remains inconsistent. MTS shows a K_d of 400 μM in amphipol, but AUC shows no interaction. The obvious explanation would be that AUC is not sufficiently sensitive to detect such weak interaction. Yet, in detergent solution a K_d of 400 μM is detectable using AUC. Because an interaction with K_d of 400 μM is “transient” regardless of the use of amphipol or detergent, the discrepancy is not properly addressed. My two cents’ worth: the MTS data cannot be used to determine a reliable K_d (because there is no indication on the plateau level), so it may be may higher than 400 μM .

The problem is exacerbated because: 1. The fits of the MTS data have huge errors, and only duplicate experiment have been done, where not even error propagation was attempted; 2. MTS was not done for the detergent sample, which makes comparison between the techniques impossible; 3. The AUC measurements, at least in panel f for L-MNG where an interaction was observed, are presented as single experiments. Especially since the signal is very small, the validity of the interpretation can be questioned without proper replicate experiments.

Additional issues regarding the tripartite complex analysis data (concerning Extended Data Figure 1):

- Additionally, the authors should also perform the MST measurements as triplicates, especially due to the large variety of measured data points (in contrast, the high-quality ITC experiments in Ex Data Fig 8 are done as triplicates!).
- Laser interferometry was used to determine the detergent concentration, however no experimental data is shown, nor the experiment is mentioned in the method part.
- FITC-labelled SiaP was used but the labelling is not mentioned in the method part

Response. As per the request by the editor, we have removed this section.

3.6 Page 5 line 20: The two-fold increased activity is very interesting – however the authors should consider that this effect might be also caused by different reconstitution efficiencies of SiaQM. The authors should provide a reconstitution efficiency or at least mention this as possible reason for the detected difference in the main text. In addition, the liposome assays are all performed in the lower activity egg-PC. Why? If the authors figured out that the protein is significantly more active in *E. coli* lipids, it might be more appropriate to perform the experiments in these lipids.

Response. We have now altered the text to acknowledge the possibility of altered reconstitution efficiency: “This increase could be a result of altered reconstitution efficiency, although analysis of the surface of SiaQM shows a number of crevices and cavities that may bind specific lipids.”

Egg-PC was used as these have been our standard model conditions to stabilise and interrogate other sialic acid transporters [Bozzola T et al ACS Chem Biol. 2022; North et al, Front Chem. 2018; Wahlgren et al Nat Commun. 2018]. We also note that model phospholipid DMPC was used for reconstitution into nanodiscs for the recently published *HiSiaQM* structure. Our functional studies started prior to structure determination, and the potential role of specific lipids was evaluated after lipids were observed in the structure. As such, the exact mechanism of how lipids influence transporter function will be the subject of further investigations (this is beyond the scope of the current study).

3.7 “Page 7 line 17: It is known since Severi et al. 2005, that SiaP is contaminated with sialic acid if the protein is expressed in LB medium (due to the casein). Protein for apo structure determination (PDB 2ECY) or analysis of the conformational transition (Glaenger et al., 2017, Biophysical Journal; Peter et al., 2021, JMB) are therefore mostly expressed in minimal medium. It is not clear if the protein prepared for ITC experiments came from minimal medium expression. In addition, the authors explain the purification of SiaP in the Method chapter with addition of 1 mM sialic acid in the SEC run, what is not compatible with subsequent substrate binding studies (the new ITC studies and the thermal shift assay experiments).

Response. This was taken in to consideration, the protein was prepared without additional sialic acid. ITC experiments were trialled with protein produced from both M9 and LB, which did not show a significant difference in K_d , indicating out our protein purification removed any contaminating sialic acid.

3.8 “Title: I would recommend changing it to a more general description, since there is not THE Trap transporter analysed in this study (maybe use “... description of a TRAP transporter”)”

Response. We have changed the title as per the editor’s recommendation.

3.9 “Page 2 line 3-5 and line 30: This is not correct, there is another class of secondary active transporters, TTTs, which share no sequence identity to TRAP transporters but also have a substrate binding protein.”

Response. We are aware of this, which is why we said “almost all other secondary transporters” on page 2 line 28.

3.10 “Page 4 line 34 and Page 5 line 1-3: I agree that the Q-subunit seems to extend the scaffold. However, the next sentence sounds like there are correlations between the lipid density and the Q-subunit scaffold properties. This can easily confuse readers, especially since there are also lipids next to the static transport domain (page 5 line 24).”

Response. We thank the reviewer for this insight. We do stand by the inference that the presence of more lipid density at the scaffold domain is related to the scaffold function of SiaQ, as mentioned in the text. As pointed out, we do see lipid density at the interface of the transport and scaffold domain, but, overall, there is more density for lipids around the scaffold domain versus the transport. We now state this in the text to avoid any confusion.

3.11 “Page 5 line 21: The *E. coli* lipid reconstitution is not explained in the corresponding method chapter (just with egg yolk phospholipids page 25 line 11).”

Response. Reconstitutions with *E. coli* phospholipids were performed similarly to with egg yolk. We have clarified this in the Methods.

3.12 “Page 5 line 26: For the whole manuscript, “sodium” is often used incorrectly to refer to “sodium ions”

Response. We have updated this throughout.

3.13 “Page 7 line 9-10: SiaP, SiaM and SiaQ are three separate peptide chains. The authors should give more details (at least in the method chapter) how they used for example AlphaFold to predict the complex. Were the peptides fused into one chain?”

Response. We have updated the methods to reflect this.

3.14 “Page 7 line 23: The thermal shift assay from extended data figure 11 should be already mentioned here for mutant R49A and E170A (I think there is a typo in E170A or should this mutant firstly mentioned in line 2 page 9?).”

Response. We have corrected this and referred now to the thermal shift data. The sentence now reads: “Mutation of predicted interacting residues at these sites validate this tripartite model—R49A (SiaP, $\alpha 2$) and D50A (SiaM, TM3a) as well as the conserved E170A (SiaP, $\alpha 5$) result in a significant reduction in transport activity (Fig. 3b) (temperature stability, sialic acid binding and purity of all SiaP and SiaQM mutants are reported in Extended Data Fig. 11).”

3.15 “Page 9 line 19 and figure 3e: It is very hard to see any upward movement in the transport domain in this representation.”

Response. This cutaway focuses on the movement of the substrate binding pocket in SiaQM relative to substrate binding pocket containing sialic acid in SiaP to suggest how the substrate might be transferred. In the figure we have increased the size of the arrow to highlight this change to the reader. We also include a supplementary Figure (Extended Data Figure 12), that better highlights the movement of the transport domain.

3.16 “Figure 1: B: sigma level is missing”

Response. Added.

3.17 “Page 24 line 22: Give the temperature for the 2h solubilization step in L-MNG. Also check the purification description of SiaQM and SiaP and add temperatures to all steps, for example lysis, affinity chromatography and SECs.”

Response. Fixed.

3.19 “Ext Data Fig 8e: The ITC measurement looks very nice, but the authors should also include the deltaH and deltaS from the triplicate measurements (only the confidence interval of the KD is mentioned in the figure description).”

Response. We thank the Reviewer, and have added deltaH and deltaS to the description with confidence intervals. We note that these values are already reported in the Figure itself.